# Experimental Ultrasound Approach for Studying Knee Intra-Articular Femur–Tibia Movements under Different Loads

**DOI:** 10.3390/jfmk9010008

**Published:** 2023-12-29

**Authors:** Ivan Ivanov, Sergey Ranchev, Stoyan Stoychev

**Affiliations:** 1National Sports Academy “Vassil Levski”, 1700 Sofia, Bulgaria; 2Institute of Mechanics, Bulgarian Academy of Sciences, 1113 Sofia, Bulgaria; serg_ran@imbm.bas.bg (S.R.); stoyan@imbm.bas.bg (S.S.)

**Keywords:** knee joint biomechanics, ultrasound scanning, femur–tibia distance

## Abstract

The purpose of the present study was to develop an experimental model for the study of intra-articular knee movements depending on the function of the knee joint and involved muscle groups under isometric stretching conditions with different loads. The experimental procedure included an ultrasound examination of a knee joint after isometric stretching in healthy men (*n* = 32). The changes (in millimeters) in the distances between the femur and tibia were measured using an ultrasound sonographer at three stages. The first stage was performed on ten (*n* = 10) healthy men in five different sitting and upright positions. In the second and third experimental model stages, lower limbs loading was applied to 22 participants. Our hypothesis, which was confirmed, was that as a result of increased loads on the participant’s back, an intra-articular decrease in the femur–tibia cartilage surface distance would be observed. The accuracy of the created experimental model was improved over its three stages from 30% to 9%. Quantitative model data can help to create a mathematical model of the mechanical effects during the deformation of knee joint bone cartilage and it can also help outline some future tasks: increasing loading weights, enlarging participant groups, performing comparisons of men and women, and performing comparisons of healthy and pathological individuals.

## 1. Introduction

The specific anatomy of each human joint determines the limits to how the quantitative parameters of the performed movements, their variety, and their characteristics can change [1,2,3]. Here, it is necessary to list the main factors affecting the mobility of the joints, such as: differences in the sizes of the contact cartilage surfaces; the volume and shape of the intra-articular cartilages; the peculiarities in the construction of the joint capsule; the number and location of the ligaments strengthening the joints; the muscle groups involved (agonists, antagonists, fixators, and neutralizers) in the moving joint; the presence of intra-articular bone synovial fluid and formations, etc.

Over 70% of the traumas that occur in the human musculoskeletal system are of joint origin, and in some specific categories of work, including sports, this percentage is significantly higher [4,5,6]. From this point of view, a joint study of intra-articular bone movements in normal and pathological individuals as a result of different loads from inner and outer forces has some benefits, as follows:Finding optimal physical exercises related to individual sportsman fitness status.Joint rehabilitation.The modeling and design of artificial joints supporting human movements.The creation of general-purpose technical devices (robots, mobile lever systems in lifting equipment, etc.).Intra-articular biomechanical processes via mathematical modeling.Information regarding intra-articular cartilage deformation under different loads.

### 1.1. Mechanical Response to Physical Loads

#### 1.1.1. Composition and Structure of Cartilage

Bone cartilage contains a layer of soft tissue providing low friction and surface load that covers the articulating bone surface in the synovial joint. It allows basic biomechanical functions to be established, such as resistance to wear, resistance to load, and shock absorption. From a biomechanical point of view, these important functional characteristics are related to the multiphasic nature of the state [7]. From an engineering point of view, porous tissue is a porous, viscoelastic material consisting of three main phases: (1) a rigid phase, which is composed primarily of a densely woven, tough, collagenous (mainly type II) fibrous network (15–22% of wet weight) covered with proteoglycan macromolecules (4–7% of wet weight). The mesh of collagen and proteoglycans represents the pores and is reinforced with fibers, forming a rigid matrix. The interstices on this porous rigid matrix are filled with water and dissolved ions. The “average” size of the pores is approximately 60 angstroms; (2) The wet phase, which is water (usually 80% of the wet weight). (3) The ionic phase, which has many types of ions—dissolved electrolytes with positive and negative charges [7,8]. These three phases react together to strengthen the tissue, which is remarkable in its ability to withstand enormous load pressures (several times that of body weight) and the high shear stresses associated with them. It is reported that stress pressure reaches up to 20 MPa in an overburdened structure [7]. The bone cartilage’s ability to withstand such high compressive loads without being torn is due to the multiphasic nature of the cartilage tissue and the unique combination of related properties of the cartilage material [9].

#### 1.1.2. Mechanical Cartilage’s Response under Physical Loading with Different Profiles

Several authors consider cartilage to be a viscoelastic material [8,10,11]. Hayes and Mockros [10] came to this conclusion on the basis of generalized Kelvin solid evaluated shear and bulk creep compliances of human articular cartilage from independent creep tests in torsion and uniaxial strains. The linearity of the compliance coefficients in the loading range tested indicated that the results are applicable to viscoelastic analyses of synovial joint mechanics. The measured compliances for normal and degenerative tissue were compared and found to differ significantly. Preliminary investigations also suggest that flow processes are not important in the initial stages of the deformation of normal tissue. Some authors consider cartilage to be a porous, viscoelastic material. For instance, Wouters et al. (2015) [11] applied Burger’s model and reported that the data revealed nonlinear relations between the applied force and the resulting deformation, with time and frequency dependence.

Cartilage, like many fine connective tissues, functions mechanically across a wide range of daily frequency loads, from <1 Hz for slow activities such as walking to 1000 Hz for high-speed activities such as jumping and impact sports [12]. Poroelasticity is known to be the basic mechanism underlying mechanical cartilage functions. Poroelasticity is manifested via friction between the synovial fluid and the hardened matrix of the cartilage and when the synovial fluid is injected under pressure into the cartilage tissue. These two elements of poroelasticity are at the basis of important mechanical cartilage functions such as self-reinforcement, energy dissipation and hydraulic permeability [12]. 

Nia et al. (2013) estimated high-frequency cartilage nanomechanics on normal cartilage and cartilage with denatured glycosaminoglycans (GAGs) [12]. The latter was due to the degradation of the cartilage matrix, which occurs in the earliest stages of osteoarthritis. In this research, high-frequency measurements that simulated a velocity under load during activities such as running and jumping were made possible by developing a combined high-frequency nanorheological system coupled to an atomic-force microscope. Through this system the authors investigated the interaction between synovial fluid and the stiffness of the cartilage matrix at the molecular level, which occurs mainly at high-frequency loads. The authors obtained some important results. The first is that GAG chains play a major role in the permeability of the synovial fluid, which is known to protect the tissue from wear and tear during activities with high load. Secondly, GAG depletion occurs in the early stages and is more vulnerable to high-frequency and rapid loading than to high-force loading. These derivatives have a direct relationship with the influence of the frequency and speed of loading on the endurance of the muscles in certain sports disciplines [12].

### 1.2. Methods for Joint Biomechanical Characterization

The wide penetration of nuclear-magnetic resonance (NMR) tests in scientific research has made it possible to evaluate some processes in the joint capsule in vivo under fixed loads. For example, Cotofana et al. found that cartilage thickness decreased to 5.2% when loading the knee with a force equal to 50% of body weight [13]. Herberthold et al. with in situ measurement of articular cartilage compression in intact femoro-patellar joints loading observed a mean in situ deformation of 44% in patellar cartilage after 3.5 h of loading (mean contact pressure 3.6 MPa) [14]. They also concluded that the femoral cartilage shows a smaller amount of deformation than the patella. They determined for the first time the amount of fluid flow from the crust into the capsular cavity. In their study, however, the load was on the foot and did not involve the participation of muscle groups. 

In another review [15], it was found that physical activity (sliding on stairs, running and climbing) leads to mild deformation of the articular cartilage, which recovers after 90 min. In another research type, studies from Kubo’s group [16,17] showed that isometric exercises increase the volume of muscle groups by up to 7%, as well as their elasticity and Young’s modulus.

Recent research [18] has reported on changes in the volume of the knee joint capsule with isometric stretching. These data convincingly show that, during active isometric exercise, there is a change in the distance between the articular surface of the large bone in the femur, which is associated with a change in the size of the muscle–tendon system of the adjacent muscle group as a result of its contraction. Additionally, Ranchev et al. [6] performed preliminary studies on the change in the kinematic chain length of the upper extremity during isometric stretching in a group of 10 men and 10 women. Their results showed that the increase in chain length reached up to 40 mm in some individuals. 

Therefore, the processes that take place inside the joint capsule can be established—the size of the change in the contact surface of the participating bone, the movement of the synovial fluid, bone cartilage deformation, etc. [19].

It is also necessary to know the specific force (in newtons) that acts on the joint. These are muscular forces and joint reactions. The magnitude and direction of the erect response greatly depend on the position of the extremity and on the acting muscle force. They cannot be calculated directly because the numbers on the unknown muscle force and reaction at the joint are much larger than the equations for equilibrium that can be written down. Therefore, in most cases, modeling and optimization methods are used, with the optimization function chosen according to the preferences of the researcher or based on physiological considerations [20]. For example, different mathematical and biomechanical models for knee joint–muscle inclusion have been obtained [21,22,23,24,25]. For the verification of the modeled results, in some of the enumerated papers surface electromyographic signals (EMGs) were used [26]. The direct relationship between these signals and the developing force is very complex and varies with different motor tasks. Nevertheless, some authors have used processed EMGs as the input excitation signal [27,28] to the so-called Hill-type model of the muscle. What is also unclear and discussed is the question of how to process EMG signals—they are low amplitude and noise-like, depending on many individual characteristics of the person being examined—gender, age, training, skin resistance, electrode placement, and others [29]. 

From what has been mentioned so far, various effects of isometric muscle work and stretching on the muscle–tendon complex can be seen. However, the influence of isometric muscle work and stretching on the functions and biomechanics of joints and the processes within them is very poorly studied. In other words, what happens inside the joint during intra-joint bone movements in normal and pathological individuals, as well as the effects of different loads (from inner and outer forces) on the joint elements of kinematics and why, as far as we know has not been investigated and clarified. 

Ultrasound echography (ultrasonography and sonography) is an approach often applied for the diagnosis of different muscle, tendon, and joint injuries [30,31,32]. The ultrasound method for the joint visualization of the lower limb is important in medicine, kinesitherapy, and biomechanics from diagnostic, therapeutic, and prophylactic prospectives [33,34,35,36,37,38]. The ease of working with ultrasound and the lack of ionizing radiation are benefits in comparison with computed tomography (CT) and fluoroscopy [39]. 

The knee joint is a complex joint and it is the most commonly injured joint because of increased vehicular trauma and sports-related injuries [40]. The complex kinematics of the knee weight-bearing position along with the complex ligamentous stability and articular congruency, are the primary reasons why these traumas are of concern to surgeons and lead to disabilities in patients [40]. 

The aim of the present study was to create an experimental in vivo ultrasonography model for evaluating the change in the distance between the bony cartilaginous surfaces of the tibia and femur inside the knee capsule under different vertical external loads during isometric muscle work of the involved muscle groups.

## 2. Materials and Methods

The presented experimental model includes an ultrasound examination of the change in the distance between the bony surfaces of the tibia and femur inside the knee capsule under different inner and vertical external loads with isometric muscle work of the applied muscle groups of healthy men (*n* = 32). The whole group was aged between 19 and 24 years (Table 1). The subjects were athletic students from the National Sports Academy “Vassil Levski”, Sofia, Bulgaria. They did not report any health problems. They completed an injury record and were informed in detail about the aim of the experiments and the procedure. All participants provided informed consent. The experimental procedure was approved by the Scientific Council of the Institute of Biophysics and Biomedical Engineering, Sofia, Bulgaria. For some of the participants, the ultrasound scans were of low quality because of knee movement and/or ultrasound transducer position changes around the knee joint center. Therefore, these data were excluded from Table 1 (empty cells) and further statistical analyses of bone-to-bone distance.

The changes (in millimeters) in the distances between the femur and tibia were measured with a portable ultrasound system VINNO 6 (Suzhou, China) with an 8–10 MHz transducer frequency, in musculo-skeletal and thyroid test mode (good intra-articular knee visualization). The RadiAnt DICOM Viewer 2022.1.1. (Poznan, Poland) was used to obtain the distances in millimeters. Statistical analysis was conducted with Sigma Plot 10 (Saint Louis, MO, USA). All ultrasound scanning was performed by the same medical physicist. 

For all experiments, the ultrasound transducer was laterally placed outside the right knee joint with the long axis coaxially oriented with the femur–tibia line (Figure 1). The transducer vision field was focused between the iliotibial band and the long-head biceps femoral tendon on the tibia (Figure 1, Figure 2, Figure 3 and Figure 4).

The model development was carried out over three stages. The first stage was performed on ten (*n* = 10) healthy men in two different sitting positions and three different upright positions—all with a knee angle between the femur and tibia of 140 degrees (Figure 2) for enhanced femur–tibia distance visualization. In two of the three upright positions, extra loads of 4 and 8 kg were vertically applied to the lower right limb to induce isometric stretching using a foot pack (Figure 2). The first stage of ultrasound testing was performed using five different lower limb poses to achieve different knee joint responses:At rest, femur–tibia angle of 140°, leg on the floor (black);Own weight stretched, femur–tibia angle of 140°, leg in the air (red);Straight, femur–tibia angle of 140°, leg without extra load (green);Straight, femur–tibia angle of 140°, leg with 4 kg extra load (yellow);Straight, femur–tibia angle of 140°, leg with 8 kg extra load (blue).

Three quantitative parameters—distance up (D_up_, femur–tibia distance nearest to the transducer measurement surface, red color in Figure 5), distance down (D_down_, femur–tibia distance in depth to the measurement transducer surface, blue color in Figure 5), and an area (A) from ultrasound pictures—were introduced (Figure 5). The two defined displacements, D_up_ and D_down_, were separated by 2.5 mm (Figure 5). The parameter D_up_ was measured for every participant in the depth of the knee joint space interval (D, mm), as shown in column 7 of Table 1. Statistical significance in comparison with Student’s t-test was determined as * *p* ≤ 0.05 and ** *p* ≤ 0.01. Student’s t-test was performed on straight, femur–tibia angle of 140°, leg without extra load (green), and two extra load positions—straight, femur–tibia angle 140°, leg with 4 kg extra load (yellow), and straight, femur–tibia angle 140°, leg with 8 kg extra load (blue).

The second (on fifteen healthy men; *n* = 15; Figure 3) and third stages (on seven healthy men; *n* = 7; Figure 4) were performed in a straight upright body positions with increasing loads on the back—0, 2, 5, 10, 15, 17, and 20 kg.

## 3. Results and Discussion

The results from stage 1 (Figure 6) show that applying extra loads statistical significantly increased D_up_ with 4 and 8 kg, D_down_ only with 8 kg, and A only with 8 kg. The obtained results for the change in the intra-articular geometry under loads can serve as a quantitative assessment of the internal joint kinematics and the determination of the individual joint mobility of the participants in the experiment [6,41,42]. 

The results obtained from stage 1 showed that the indetermination (accuracy or error, %) in obtaining the distances in millimeters between the femur and the tibia in the knee joint was high. The reason for this was mainly the inaccuracy of the ultrasound transducer positioning around the knee joint for the same participant despite the use of a marker outline for the ultrasound transducer on the knee joint skin. The authors concluded that the reproducibility of the obtained ultrasound pictures was poor. For this reason, we decided to minimize the indetermination (inaccuracy) by testing the same distance in the knee joint but in a straight posture with increasing loads on the back (Figure 3) to induce a lower limb shortening. Some of the results of this second stage have been published [41,42]. 

The second stage was performed on fifteen (*n* = 15) healthy men in a straight upright body position with increasing loads on the back—0, 2, 5, 10, 15, 17, and 20 kg [41] (Figure 3).

Our hypothesis, which was confirmed, was that, as a result of increasing extra loads on the participant’s backs (2, 5, 10, 15, 17 and 20 kg), an intra-articular decrease in the femur–tibia cartilage surface distance would be observed during static straight poses. This change in the distance between the bones indicates the presence of intra-articular processes, which in turn depends on the following factors:The extra load levels.The biomechanical properties of the knee joint components—femur, tibia, fibula, and patella cartilage deformability, knee joint ligament and tendon viscoelasticity, considering that these are unique to each person and depend on various factors (age, gender, height, level of training, etc.).Amount and viscosity of synovial fluid.The lower limb pose when increasing extra load.Age, sex, and weight.

The regression values for the femur–tibia distances D for the twenty-two tested participants in stages 2 and 3 were determined from the individual linear regression equations for all subjects. All collected data fell within the 95% confidence interval around the regression line. The percentage relative reduction in femur–tibia distances was calculated based on the obtained regression model with individual angle coefficients (Figure 7, Table 1).

The third stage was carried out to further the model optimization by increasing its accuracy. For this purpose, an ultrasound transducer muff was made to immobilize the transducer on the knee joint (Figure 4). The results obtained in the third stage are shown in Table 1 (N26–N32). The mean percentage decrease for these seven participants (column % decrease in Table 1) is smaller than in participants N11–N25. This is an indirect measure for the aimed model optimization.

The results obtained over the three stages for the change in the intra-articular geometry under different loads and stretching serve as a quantitative assessment of the internal joint kinematics and the determination of the individual joint mobility of the participants in the experiment [6,41,42]. 

The accuracy of our measurements in the proposed experimental model, using the VINNO 6 device is limited by three components. The first is related to the used a transducer’s accuracy characteristics. The second is dependent on the accuracy in the identity of the reproduction of the transducer-–knee joint image position. The third is determined by the researcher’s skill in terms of pictures scans, treatment, and obtaining the distances in millimeters.

The first accuracy component is lower than 5% and is defined and described in the VINNO 6 user manual for our concrete transducer type (F4-12L) and used experimental mode. This accuracy level was the same for all three stages (Table 2). 

The second accuracy component was different for the different model stages. For the first stage this second component reached 20%. In the second stage this component decreased to 7%, due to strict adherence to the identity of the transducer-knee joint image position reproduction ensuring maximal ultrasound transducer positioning in relation to the knee joint for each participant.

In the third stage the model accuracy decreased to 2%. The third accuracy component was minimized to 2% due to the fact that all ultrasound picture scans were made using an ultrasound transducer muff to immobilize the transducer on the knee joint.

Currently, the present experimental model’s accuracy was defined as the sum of the three described components and is lower than 9% (Table 2). 

## 4. Limitations of This Study

There were several limitations of the present study. Firstly, there is a methodological limitation related to ultrasound transducer positioning around the knee joint for the same participant, despite the use of a marker outline for the ultrasound transducer on the knee joint skin. Although we made several efforts to minimize these methodological limitations, the final approach accuracy remained high at 9% [37]. Secondly, the tested group of participants were only males, with different movement habits and fitness statuses. As a result, they showed femur–tibia displacements, which were appropriate only for model verification. In future investigations, females and regular sportsman must be included. Thirdly, the maximal used extra loads were 20 kg on compression. The reasons for this were related to preserving participants’ joints and ensuring body protection. These extra loading levels will be increased in the future as a percentage of body weight in order to emphasize the observed processes. Finally, the tested group had only 32 males divided into three experimental model stages (*n*1 = 10; *n*2 = 15; *n*3 = 7). This relatively small participant number decreases the validity of the model findings and the importance of the obtained scientific conclusions. Тhe analyzed limitations of this study show directions for future tasks related to improving the model’s accuracy and increasing its scientific value [33,38,43,44,45,46,47]. 

## 5. Conclusions

To the best of our knowledge, our experimental model is the first to investigate the reduction in the distance between the femur and the tibia in the knee joint at 0° flexion with increased load. In this paper, the authors summarized and clearly described all stages of experimental model development with the obtained quantitative results. The differences from previous studies are as follows:A first-time description of the experimental model stage 1 data with graphs.A first-time description of the model’s final accuracy, as shown in Table 2.First time data separation for stages 2 and 3.

The commonalities with our previous studies are as follows:A description of the experimental model data, as shown in Table 1 (stages 2 and 3).

The presented findings of the ultrasound model will be the basis for future investigations: (1) the determination of the change in the contact area between the femur and the tibia under different axial loads; (2) an evaluation of the deformation of the cartilage tissue from the contact area between the femur and the tibia under different axial loads; and (3) modeling the interaction between cartilage deformation and interstitial fluid flow from the cartilage into the joint cavity under loading conditions. The difficulties of the method in accurately measuring femur–tibia distance are, firstly, related to ensuring the immobility of the ultrasound transducer on the knee joint and, secondly, minimizing small movements in the knee of the tested participants.

In the global literature, there is information that the contact surface between the femur and the tibia varies between 2 cm^2^ and 6 cm^2^ and this is influenced by the menisci [48,49]. Some authors concluded that the menisci may occupy 70% of the total femur–tibia contact area [50].

The obtained quantitative data for femur–tibia distances combined with the femur–tibia contact surface area will contribute to the development of a future mathematical model for mechanical effects during the deformation of the knee joint femur and tibia cartilages. Additionally, based on these data, we will attempt to prepare a quantitative method with a software program to automatically calculate femur-tibia kinematics from ultrasound images. 

On the basis of our experimental model, one can outline some future tasks: Increase loading weight.Enlarge participant groups.Compare men and women.A study of cartilage deformation under stretch loading.Identify the influence of isometric stretching on knee hemorheology [51].The development/design of exercises in order to separately divide and estimate the contribution to the load of the joint cavity from stretching only and in combination with other loads.The development of practically applicable mechanical trainers for joint fitness.The creation of an experimental in vivo ultrasonography model for the evaluation of the change in the distance between the bony cartilaginous surfaces inside the ankle joint under different vertical external loads. 

## Figures and Tables

**Figure 1 jfmk-09-00008-f001:**
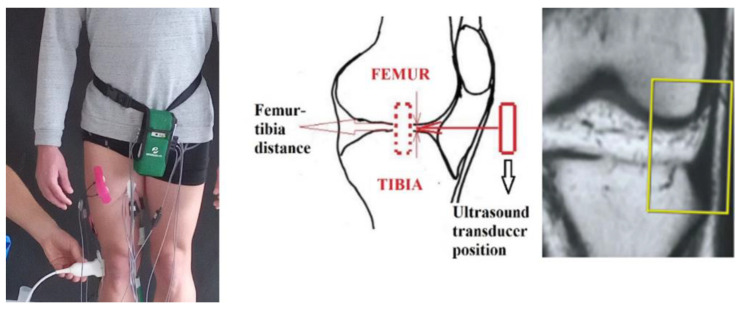
Ultrasound transducer position on the knee joint. The yellow box shows the ultrasound transducer viewpoint.

**Figure 2 jfmk-09-00008-f002:**
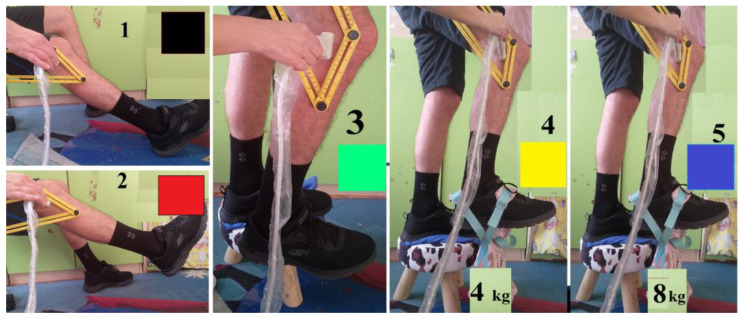
Stage 1 experimental setup showing lower limb poses with loads. The colors show the 5 different lower limb poses used to achieve different knee joint responses.

**Figure 3 jfmk-09-00008-f003:**
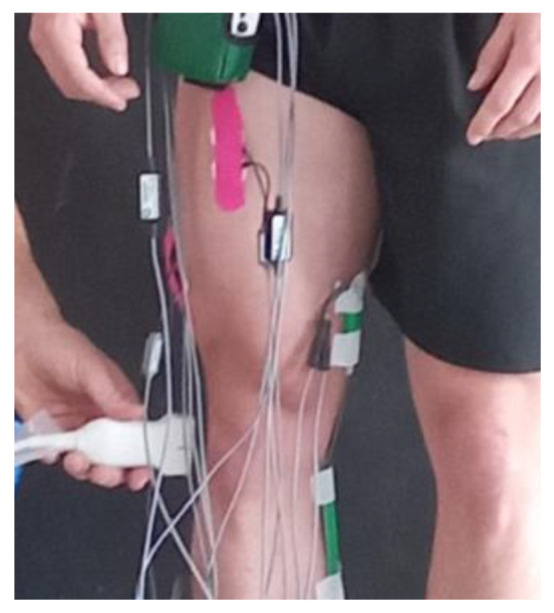
Ultrasound transducer position on knee joint in stage 2 [41].

**Figure 4 jfmk-09-00008-f004:**
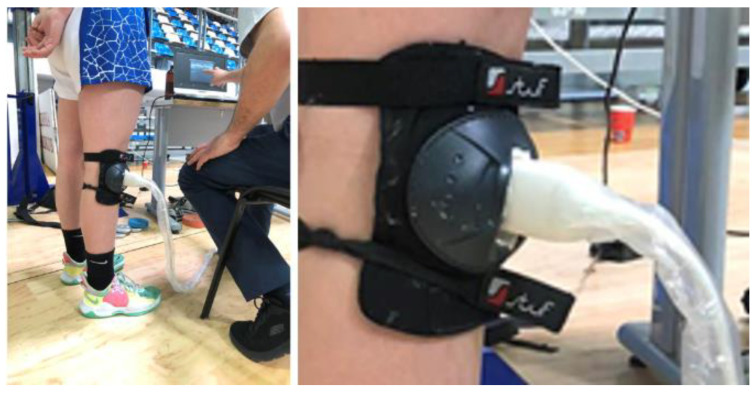
Experimental setup in stage 3, with ultrasound transducer muff to immobilize the transducer on the knee joint.

**Figure 5 jfmk-09-00008-f005:**
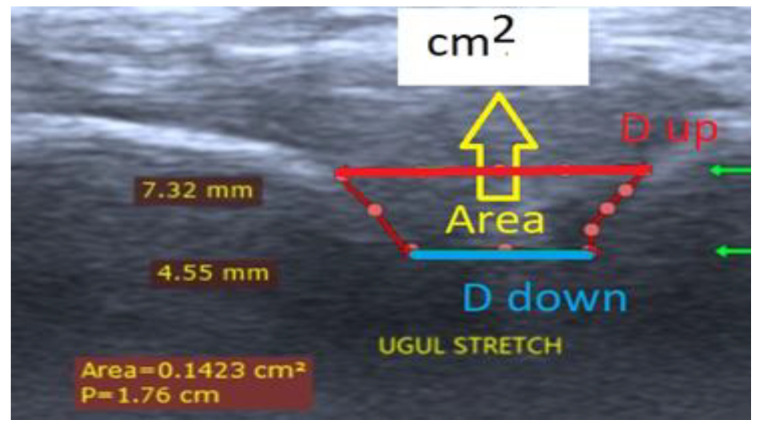
A screen view of the VINNO 6 echograph with the measured distances between the femur and tibia bones in the knee joint for one participant. Reproduced from [42]. D_up_, femur–tibia distance nearest to the transducer measurement surface, red color; D_down_—femur–tibia distance in depth to the measurement transducer surface, blue color; area A—between D_up_, D_down_ and bone surfaces.

**Figure 6 jfmk-09-00008-f006:**
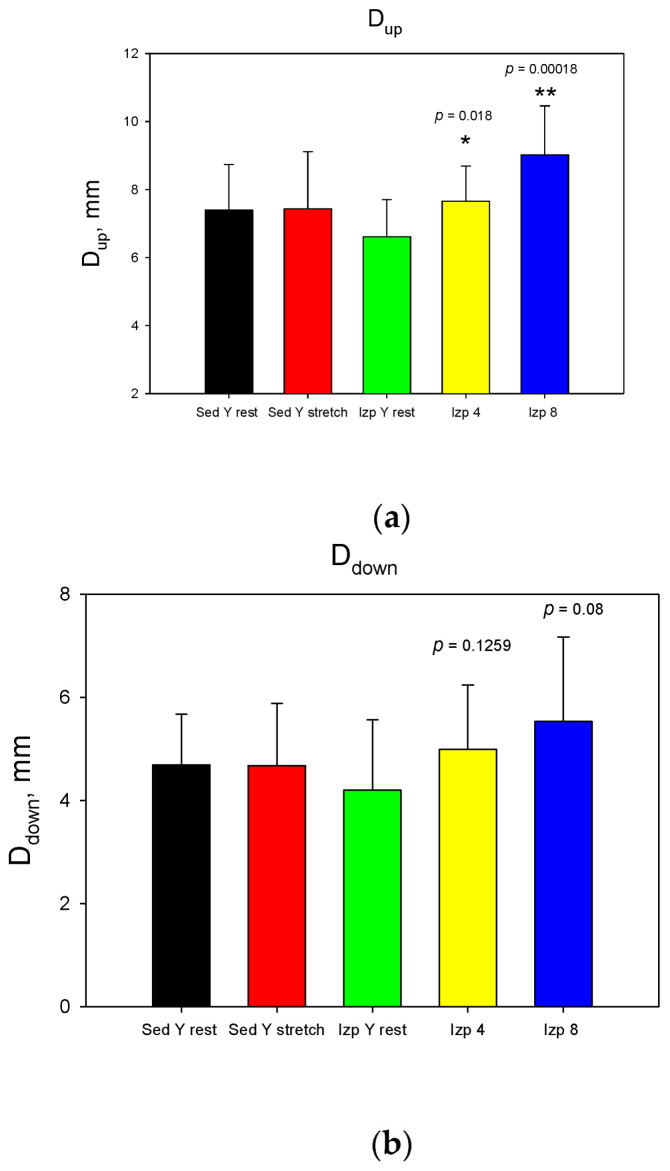
Data comparison for the first stage: (**a**) D_up_; (**b**) D_down_; (**c**) area. The statistical significance determined with Student’s t-test was shown as * *p* ≤ 0.05, ** *p* ≤ 0.01. The colors correspond to those used in Figure 2.

**Figure 7 jfmk-09-00008-f007:**
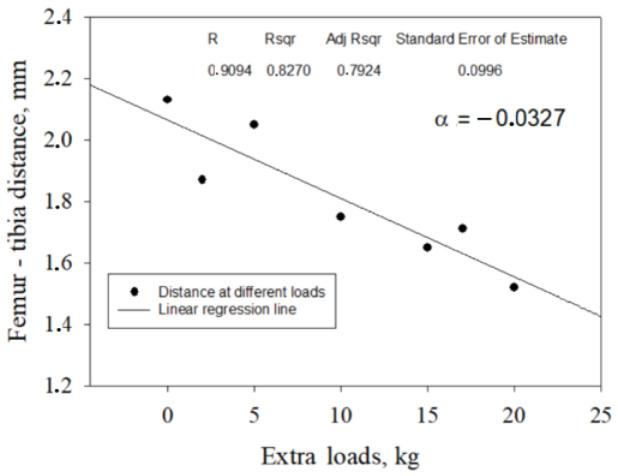
Femur–tibia distance versus extra loads for participant N14, with the highest femur–tibia distance decrease at 30.35%, with the highest weight recorded at 103 kg.

**Table 1 jfmk-09-00008-t001:** Results for anthropometry, BMI, BSA, femur–tibia distances and additional data for all participants in stages 2 and 3.

Participants						D, mm	0 kg	2 kg	5 kg	10 kg	15 kg	17 kg	20 kg	Angle Coefficient	%, Decrease	Stage
Number	Age	Height, cm	Weight, kg	BMI (kg/m^2^)	BSA (m^2^)											
N11	20	175	70	22.86	1.85	1	1.5	1.58	1.3	1.39	1.26	1.23	1.36	−0.0117	15.74	II
N12	22	171	70	23.94	1.82	1	1.65	1.57	1.6	1.55	1.46	1.7	1.37	−0.0072	8.81	
N13	21	176	77	24.86	1.93	1	1.28	1.24	1.14	1.13	1.09	1.06	1.02	−0.0116	18.55	
N14	22	188	103	29.14	2.29	1.5	2.21	1.9	2.05	2.06	1.53	1.46	1.6	−0.0327	30.35	
N15	20	183	76	22.69	1.98	1	1.45	1.02	1.09	1.16	0.996	1.03	1.05	−0.0119	19.18	
N16	20	181	86	26.25	2.07		2.05	2	1.85		1.8	1.78	1.71	−0.0147	14.61	
N17	20	183	90	26.87	2.12	1	1.78	1.54	1.52	1.47	1.62	1.71	1.49	−0.0032	3.94	
N18	19					1	0.895	0.651		0.673	0.456	0.439		−0.022	53.92	
N19	19	170	78	26.99	1.89	1	1.41	1.36	1.4	1.32	1.25	1.19	1.17	−0.0121	17.08	
N20	20	177	96	30.64	2.13	1.25	1.66	1.58	1.51	1.47	1.57	1.38	1.37	−0.0112	13.89	
N21	22	175	64	20.90	1.78	0.75	1.62	1.44	1.51	1.52		1.44		−0.0094	11.76	
N22	20	190	80	22.16	2.08	0.9	1.80		1.71	1.57	1.55	1.56	1.54	−0.0132	14.9	
N23	20	185	90	26.30	2.14	1	2.13	1.87	2.05	1.75	1.65	1.71	1.52	−0.0256	24.78	
N24		191	85	23.30	2.14	1	1.84						1.39	−0.0225	24.46	
N25	21	167	70	25.10	1.79	1.25	1.93	1.82		1.76		1.54	1.41	−0.0233	24.27	
N26	22	184	73	21.56	1.95	0.875	1.09	1.08	1.07	1.03	0.979		0.99	−0.0059	10.737	III
N27	20	184	69	20.38	1.90	1.5	0.595						0.489	−0.0053	17.81	
N28	21	183	86			1	1.25	1.22	1.2	1.25	1.21		1.18	−0.0022	3.54	
N29	24	185	79	23.08	2.03	1	1.86	1.8	1.77	1.75	1.77			−0.0052	5.7	
N30	20	193	79	21.21	2.09	1.125	1.84	1.8	1.89	1.66	1.61		1.61	−0.0139	15	
N31	24	178	80	25.25	1.98	1	1.19	1.18	1.14	1.05	1.07		1.08	−0.0064	10.92	
N32	19	187	74	21.16	1.99	1.5	0.648		0.656	0.629	0.619			−0.0684	6.87	

**Table 2 jfmk-09-00008-t002:** Experimental model accuracy.

Stage/Component	1	2	3	Total
Stage 1	≤5%	≤20%	≤5%	≤30%
Stage 2	≤5%	≤7%	≤5%	≤17%
Stage 3	≤5%	≤2%	≤2%	≤9%

## Data Availability

The data presented in this study partly are openly available in
Figure 1
at [doi: 10.3390/app13158596] and in
Table 4
at [doi: 10.7546/ijba.2023.27.2.000946].

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
