# Peer review of "Experimental Ultrasound Approach for Studying Knee Intra-Articular Femur–Tibia Movements under Different Loads"

_jfmk, 2023, doi:10.3390/jfmk9010008_

Round 1
Reviewer 1 Report
Comments and Suggestions for Authors
See attachment

Comments on the Quality of English LanguageSee attachment
Author Response
Reviewer 1 answer:
Generally, the article is well written with good presentation of main problem and good description of
own research.
Ultrasound method is used from several years. But looking at the presented photograph one can have
doubt if such accuracy up to 0.1 mm can be obtained without an error.
The load steps of 2, 5, 10, 15, 17, 20 kg are presented within the Abstract and then within the Results
and Discussion. It should be given in Material and Methods.
It was added text in Material and Methods: 215 – 217. “The second (on fifteen healthy men, n=15) (fig. 3) and the third stage (on seven healthy men, n=7) (fig. 4) was performed at straight upright body position with increasing of loads on the back – 0, 2, 5, 10, 15, 17 and 20 kg.”
Within the article I have found the following editorial errors:
Lines:
53: weight). , → weight), Corrected.
156: benefts → benefits Corrected.
157: fluoroscopy → fluoroscopy - Corrected.
163-171: take it to the Introduction, after 157 - Corrected.
175: of 32 healthy men (n=32) → of healthy men (n = 32) [no repetition] - Corrected.
216: throw → through - Corrected.
223: deapth → depth - Corrected.
226: showen → shown - Corrected.
216-226: take it to the Material and Methods [some parts of the text are repetitions] - Corrected.
232: inertial joint kinematics → inertial joint kinetics [kinematics deals only with displacement, time,
frequency, velocity, acceleration] - Corrected.
250: a intra → an intra - Corrected.
251: will → would be - Corrected.
260: limp → limb - Corrected.
266: coeficient → coefficient - Corrected.
281: → of the our → of our - Corrected.
321: model can outline → model one can outline - Corrected.
The Authors thank a lot to the Reviewer 1 recommendations. They were forced with time for the manuscript submitting and they answer to the Reviewers very short first time. They apologize for it.

Reviewer 2 Report
Comments and Suggestions for Authors
Introduction - 1 and 2 paragraphs: reference should be presented. – Ref. No. 1 is not enough to explain
Line 163-171: This content is more appropriate to the introduction than the research method. It's a good idea to delete it.
Figure 5: What is the difference between the left and right figures? If a simple magnification of the picture on the left is the picture on the right, it is better to unify and present a clearer picture.
Table 1. What does the empty information in N18, N24, and N28 mean? Isn't it right to be excluded from the subject if you haven't measured it?
Discussion & Conclusion:
What the results of this study mean must be clearly presented. Also, differences and commonalities from previous studies should be presented.
Furthermore, the limitations of this study that the authors think should be described.
The use of commas(,) and dots(.) should be distinguished throughout the paper.
Author Response
Dear Reviewer,
many thanks for all suggested corrections. Absolutely all of your suggestions have
been taken into account in the revised manuscript. Thanks for your kindness and help in improving the quality of our article.
Round 2
Reviewer 2 Report
Comments and Suggestions for Authors
Authors have not been able to give appropriate answers to reviewers' questions. All answers should be in a point-to-point format, but the explanation is insufficient.
In addition, the limitations of this study, which the authors revised and presented, mentioned the fundamental problem of experimental design, which was an important part to be considered from the research design stage.
Moreover, it is difficult to clearly understand what the results of this study mean. It was asked to clarify the meaning of this study by presenting the differences and commonalities from previous studies, but it is difficult to find such corrections, and the authors are unable to explain them.
Author Response
Answer Reviewer 2 first letter.
Introduction - 1 and 2 paragraphs: reference should be presented. – Ref.
No. 1 is not enough to explain.
A new five references were added to Introduction between row 23 and row 36.
Line 163-171: This content is more appropriate to the introduction than
the research method. It's a good idea to delete it.
The text between Line 163-171 was reduced to two sentences and they were add to Introduction.
Figure 5: What is the difference between the left and right figures? If a simple magnification of
the picture on the left is the picture on the right, it is better to unify and present a clearer picture.
The left picture was deleted.
Table 1. What does the empty information in N18, N24, and N28 mean?
Isn't it right to be excluded from the subject if you haven't measured it?
Between the rows 177 – 180 it was described “For some of the participants, the ultrasound scans were with low quality because of knee movement and/or ultrasound transducer position changes around the knee joint center. Therefore, these data were excluded from Table 1 (empty cells) and from further statistical analyses of bone-to-bone distance.”
Discussion & Conclusion:
What the results of this study mean must be clearly presented. Also, differences and commonalities from previous studies should be presented.
Rows 319 – 329: “To the best of our knowledge, our experimental model is the first for investigation the reduction of the distance between the femur and the tibia in the knee joint at 0° flexion with increased load. In this paper the authors summarize and clearly described all stages for experimental model development with obtained quantitative results. The differences from the previous studies are:
- first time description of the experimental model stage 1 data with graphs;
- first time description of the model final accuracy in Table 2;
- first time data separation for stage two and stage three.
The commonalities with our previous studies are:
- description of the experimental model stages two and three data in Table 1.
Presented ultrasound model findings will be the basis for the future investigations: (1) determination of the change in the contact area between the femur and the tibia under different axial loads; (2) evaluation of the deformation of the cartilage tissue from the con-tact area between the femur and the tibia under different axial loads; (3) modelling the in-teraction between cartilage deformation and interstitial fluid flow from the cartilage into the joint cavity under loading conditions. The difficulties of the method for accurate meas-urement of the femur-tibia distance are related to ensuring the immobility of the ultra-sound transducer regarding the knee joint and second, minimizing small movements in the knee of the tested participants.”.
Furthermore, the limitations of this study that the authors think should be described.
- Limitations of the study.
There are several limitations of the present study. Firstly, we have a methodological limitation related to ultrasound transducer positioning about the knee joint for the same participant, despite of the used marker outline for ultrasound transducer on the knee joint skin. Although we have made several eforts to minimize this methodological limitations, the final approach accuracy stay high – 9%. [46]. Secondly, the tested group of participants were only males with different movement habits and fitness status. As a result, they showed femur-tibia displacements which were appropriate only for model verification. In future investigations it was needed to include females and regular sportsman. Third, the maximal used extraloads were 20 kg on compression. The reasons were related to save participants joint and body protection. This extraloading levels will be increased in the future as percent of body weight in order to be emphasized the observed processes. Fourtly, the tested group were with only 32 males divided to the three experimental model stages (n1=10; n2=15; n3=7). This relatively small participant number decrease the model findings depth and the obtained scientific conclusion importance. Тhe analyzed limitations of the study show the directions for future tasks related to improving the model accuracy and increasing its scientific value [47 – 53].
The use of commas(,) and dots(.) should be distinguished throughout the paper.
The use of commas(,) and dots(.) were distinguished throughout the paper by Word check spelling.
Answer Reviewer 1 second letter.
Authors have not been able to give appropriate answers to reviewers' questions. All answers should be in a point-to-point format, but the explanation is insufficient.
The Authors give the answer point-to-point format now. They believe that this is an appropriate answer.
In addition, the limitations of this study, which the authors revised and presented, mentioned the fundamental problem of experimental design, which was an important part to be considered from the research design stage.
Moreover, it is difficult to clearly understand what the results of this study mean. It was asked to clarify the meaning of this study by presenting the differences and commonalities from previous studies, but it is difficult to find such corrections, and the authors are unable to explain them.
Line 163 – 166: The aim of the present study was to create an experimental in vivo ultrasonographycal model for evaluation the change in the distance between the bony cartilaginous surfaces of the tibia and femur inside the knee capsule under different vertical external loads with isometric muscle work of applied muscle groups.
The Authors correct the final Conclusion in relation to the Reviewer 1 recommendations: “To the best of our knowledge, our experimental model is the first for investigation the reduction of the distance between the femur and the tibia in the knee joint at 0° flexion with increased load. In this paper the authors summarize and clearly described all stages for experimental model development with obtained quantitative results. The differences from the previous studies are:
- first time description of the experimental model stage 1 data with graphs;
- first time description of the model final accuracy in Table 2.
The commonalities with our previous studies are:
- description of the experimental model stages two and three data in Table 1.
Presented ultrasound model findings will be the basis for the future investigations: (1) determination of the change in the contact area between the femur and the tibia under different axial loads; (2) evaluation of the deformation of the cartilage tissue from the con-tact area between the femur and the tibia under different axial loads; (3) modelling the in-teraction between cartilage deformation and interstitial fluid flow from the cartilage into the joint cavity under loading conditions. The difficulties of the method for accurate meas-urement of the femur-tibia distance are related to ensuring the immobility of the ultra-sound transducer regarding the knee joint and second, minimizing small movements in the knee of the tested participants.”.
The Authors thank a lot to the Reviewer 2 recommendations. They were forced with time for the manuscript submitting and they answer to the Reviewers very short first time. They apologize for it.
Answer Reviewer 2 first letter.
Introduction - 1 and 2 paragraphs: reference should be presented. – Ref.
No. 1 is not enough to explain.
A new five references were added to Introduction between row 23 and row 36.
Line 163-171: This content is more appropriate to the introduction than
the research method. It's a good idea to delete it.
The text between Line 163-171 was reduced to two sentences and they were add to Introduction.
Figure 5: What is the difference between the left and right figures? If a simple magnification of
the picture on the left is the picture on the right, it is better to unify and present a clearer picture.
The left picture was deleted.
Table 1. What does the empty information in N18, N24, and N28 mean?
Isn't it right to be excluded from the subject if you haven't measured it?
Between the rows 177 – 180 it was described “For some of the participants, the ultrasound scans were with low quality because of knee movement and/or ultrasound transducer position changes around the knee joint center. Therefore, these data were excluded from Table 1 (empty cells) and from further statistical analyses of bone-to-bone distance.”
Discussion & Conclusion:
What the results of this study mean must be clearly presented. Also, differences and commonalities
from previous studies should be presented.
Rows 319 – 329: “To the best of our knowledge, our experimental model is the first for investigation the reduction of the distance between the femur and the tibia in the knee joint at 0° flexion with increased load. In this paper the authors summarize and clearly described all stages for experimental model development with obtained quantitative results. The differences from the previous studies are:
- first time description of the experimental model stage 1 data with graphs;
- first time description of the model final accuracy in Table 2;
- first time data separation for stage two and stage three.
The commonalities with our previous studies are:
- description of the experimental model stages two and three data in Table 1.
Presented ultrasound model findings will be the basis for the future investigations: (1) determination of the change in the contact area between the femur and the tibia under different axial loads; (2) evaluation of the deformation of the cartilage tissue from the con-tact area between the femur and the tibia under different axial loads; (3) modelling the in-teraction between cartilage deformation and interstitial fluid flow from the cartilage into the joint cavity under loading conditions. The difficulties of the method for accurate meas-urement of the femur-tibia distance are related to ensuring the immobility of the ultra-sound transducer regarding the knee joint and second, minimizing small movements in the knee of the tested participants.”.
Furthermore, the limitations of this study that the authors think should be described.
- Limitations of the study.
There are several limitations of the present study. Firstly, we have a methodological limitation related to ultrasound transducer positioning about the knee joint for the same participant, despite of the used marker outline for ultrasound transducer on the knee joint skin. Although we have made several eforts to minimize this methodological limitations, the final approach accuracy stay high – 9%. [46]. Secondly, the tested group of participants were only males with different movement habits and fitness status. As a result, they showed femur-tibia displacements which were appropriate only for model verification. In future investigations it was needed to include females and regular sportsman. Third, the maximal used extraloads were 20 kg on compression. The reasons were related to save participants joint and body protection. This extraloading levels will be increased in the future as percent of body weight in order to be emphasized the observed processes. Fourtly, the tested group were with only 32 males divided to the three experimental model stages (n1=10; n2=15; n3=7). This relatively small participant number decrease the model findings depth and the obtained scientific conclusion importance. Тhe analyzed limitations of the study show the directions for future tasks related to improving the model accuracy and increasing its scientific value [47 – 53].
The use of commas(,) and dots(.) should be distinguished throughout the paper.
The use of commas(,) and dots(.) were distinguished throughout the paper by Word check spelling.
Answer Reviewer 1 second letter.
Authors have not been able to give appropriate answers to reviewers' questions. All answers should be in a point-to-point format, but the explanation is insufficient.
The Authors give the answer point-to-point format now. They believe that this is an appropriate answer.
In addition, the limitations of this study, which the authors revised and presented, mentioned the fundamental problem of experimental design, which was an important part to be considered from the research design stage.
Moreover, it is difficult to clearly understand what the results of this study mean. It was asked to clarify the meaning of this study by presenting the differences and commonalities from previous studies, but it is difficult to find such corrections, and the authors are unable to explain them.
Line 163 – 166: The aim of the present study was to create an experimental in vivo ultrasonographycal model for evaluation the change in the distance between the bony cartilaginous surfaces of the tibia and femur inside the knee capsule under different vertical external loads with isometric muscle work of applied muscle groups.
The Authors correct the final Conclusion in relation to the Reviewer 1 recommendations: “To the best of our knowledge, our experimental model is the first for investigation the reduction of the distance between the femur and the tibia in the knee joint at 0° flexion with increased load. In this paper the authors summarize and clearly described all stages for experimental model development with obtained quantitative results. The differences from the previous studies are:
- first time description of the experimental model stage 1 data with graphs;
- first time description of the model final accuracy in Table 2.
The commonalities with our previous studies are:
- description of the experimental model stages two and three data in Table 1.
Presented ultrasound model findings will be the basis for the future investigations: (1) determination of the change in the contact area between the femur and the tibia under different axial loads; (2) evaluation of the deformation of the cartilage tissue from the con-tact area between the femur and the tibia under different axial loads; (3) modelling the interaction between cartilage deformation and interstitial fluid flow from the cartilage into the joint cavity under loading conditions. The difficulties of the method for accurate measurement of the femur-tibia distance are related to ensuring the immobility of the ultra-sound transducer regarding the knee joint and second, minimizing small movements in the knee of the tested participants.”.
The Authors thank a lot to the Reviewer 2 recommendations. They were forced with time for the manuscript submitting and they answer to the Reviewers very short first time. They apologize for it.

Round 3
Reviewer 2 Report
Comments and Suggestions for Authors
The authors responded appropriately.